# "Being Diverse and Being Included, Don't Go Together in Policing"—Diversity, Inclusion, and Australian Constables

Toby Miles-Johnson [1,*] and Suzanna Fay [2]

1    School of Social Sciences, Western Sydney University, Liverpool, NSW 2710, Australia
2    School of Social Science, University of Queensland, St. Lucia, QLD 4072, Australia; s.fay@uq.edu.au
*    Correspondence: t.miles-johnson@westernsydney.edu.au

**Abstract:** Across the globe, there is little research that examines the impact of diversity on police practice, particularly whether it increases or decreases the competency of the police organization or whether police officers perceive diversity within the organization and the addition of diverse officers as positive or negative. Contributing new findings to the extant policing literature, this research analyzes data collected from interviews with forty-six constables working in one of the largest Australian state police organizations. Contributing five key findings regarding diversity and inclusion in policing, this research suggests that lack of acceptance of diversity broadly, and bias towards diverse identified officers, results in the exclusion of officers, and a workforce that is fragmented. The lack of unification constables in this research have with diverse colleagues is concerning given that a cohesive police team increases the safety of all officers, improves the effectiveness of police response, strengthens the communication between police and citizens (as well as communication within the organization), increases the morale of officers, and will support the legitimacy of the organization. Whilst constables in this study were not asked questions about their own implicit or explicit levels of bias towards members of diverse groups, the unsolicited responses from many of the constables, as well as the recognition of Whiteness in terms of the racial identity of many officers within the organization, suggests that constables in this study are biased towards officers that are not part of the majority group.

**Keywords:** police; diversity; inclusion; identity; bias; whiteness

## 1. Introduction

Previous research suggests that police organizations across the globe are strategically diversifying the workplace and the personnel of its officers, and diversification has become a priority underpinning police recruitment and workplace practices (see [1–3]). Like other global police agencies, Australian police organizations realized the need for diversification and since the 1990s purposefully sought to recruit people from diverse groups [4–6]. Supported by changes to workplace legislation and legal changes regarding diversity initiatives, Australian police organizations slowly increased the employment of people historically underrepresented as police officers [5,7]. This included increasing the number of diverse employees based on observable characteristics, such as gender, race, ethnicity, and age, as well as less observable characteristics such as sexuality, ability, education level, and religion [8].

Creating workplace policies to support diversity and inclusion of officers from different groups, all Australian police organizations created liaison officer roles in a bid to attract diverse people into policing, as well as targeting specific recruitment drives to attract diverse people into general-duties police work [9,10]. Whilst workplace policies were created to support these initiatives, and were deemed successful by Australian police organizations, the number of officers employed from diverse groups is far less than the number of officers employed from majority groups [3,11]. This is not to suggest that

Australian police organizations are not diversifying in terms of their personnel and the number of officers working within general duties, but officers identifying as members of the majority group outweigh the number of officers identifying as a member of a diverse group [8]. For example, like policing in the United States, United Kingdom, and Canada, Australian police organizations employ more male than female officers, most male and female officers identify as heterosexual, most police organizations comprise more White officers than officers from other racial groups, and most officers who identify as religious, categorize themselves as being part of a majority religion such as being either Catholic or Christian [12–14].

There is little research, however, which examines the impact of diversity on police organizations around the globe, particularly whether it increases or decreases the competency of the police organization or whether police officers perceive diversity within the organization and the addition of diverse officers as positive or negative. This is surprising given that most police organizations initiated strategic plans regarding diversity and inclusion practices. Much of the research examining diversity and inclusion in policing focuses on policing as a service and how it can improve community policing and citizen engagement [15]. It also focuses on discrimination in the workplace and career prospects of officers from diverse groups but does not examine diversity and inclusion from the perspective of officers and how diversity and inclusion affect police practice.

Increasing the diversity of police organizations is an ongoing endeavor, and one that is a priority for many Australian police organizations [7,8]. A police organization comprising equal numbers of diverse officers and officers from the majority group is more likely to understand all the citizens it polices, as well as have a more cohesive and representative force [8,16–18]. Whilst Australian police organizations have embraced this idea and published recruitment guidelines, which support the idea of all citizens being able to undertake a career in policing, many organizations are criticized for discriminatory recruitment practices [3,8,19]. Much of the criticism focuses on biased workplace practices and those which identify bias towards members of diverse groups regarding capability and competency [3,8,20].

It is argued that organizations such as the police should target recruitment drives to appeal to specific groups thereby increasing the representation of all community members [20–22]. Targeted recruitment drives, however, are not always successful, with recruitment initiatives falling short in increasing the number of diverse recruits [23]. This is because people from diverse communities often have poor community relations with police, negative community history regarding police–citizen encounters, and are hesitant to join an organization that is perceived to represent (and comprise) majority group members [8,11,23]. Whilst targeted recruitment of diverse people into policing may decrease inequalities and increase fair workplace practices for all officers, it is also likely that majority group officers within an organization could become sensitized to the concerns of diverse people through increased contact as colleagues [7]. Yet this is a research topic that needs systematic inquiry, particularly in an Australian context.

As stated, police organizational culture in Australia is moving forward in terms of the value it places on diversity within the workforce and Australian police organizations are aware of the potential impact this may have on police–citizen engagement and police work relating to service delivery [8]. Critics argue, however, that changes regarding recruitment of diverse people in policing are not happening quickly enough and some of the policies regarding recruitment (such as 50 percent initiatives regarding male and female numbers or recruitment of lesbian, gay, bisexual, transgender, intersex, and queer people) unfairly target some groups and create tensions between groups from the onset of training [11,13,14,24]. Whilst bias (explicit and implicit) and negative stereotypes regarding members of a police organization's workforce can occur across all levels of an organization, it is argued that it is exacerbated when bias stems from top-down practices consciously or unconsciously supported by management or senior officers [25].

Managerial bias has been shown to strongly influence employee behavior and has a negative impact on the collegiality of employees [7,22]. Heavily criticized as being organizations entrenched in bias, police organizations are affected by top-down practices, and previous research has examined the impact this has on police workplace practices and the ability of officers to work cohesively and collegially [26]. However, the body of this research has not examined the impact of diversity and inclusion on police practice and mainly focused on workplace interaction and interpersonal treatment and how these change over time [22]. Whilst research suggests that police officers' attitudes towards work are shaped by police culture, time in the job, organizational policies, and practices, positive and or negative interactions with colleagues, and conscious and unconscious biases and stereotypes (see [14,27–30]), it is unknown whether diversity and inclusion have an impact on internal workplace practices in Australian policing. This is a distinct gap in knowledge and one that needs further research. This research, therefore, sought to address this gap in knowledge regarding diversity and inclusion, and the effect they have on police practice.

## 2. Materials and Methods

### 2.1. Interviews with Constables

A rigorous qualitative methodological approach was applied during the completion of this research. This included conducting semi-structured interviews with 46 constables in one Australian state to better understand how diversity and inclusion affect police practice. Interviews were conducted within a 12-month period, and this included interviews with constables working in the capital city and regional areas. Twenty-two discussion prompts regarding diversity and inclusion were used to facilitate the semi-structured interviews (see Appendix A). The final sample comprised 24 male constables and 22 female constables. In accordance with research stipulations outlined by the police organization, no senior officers ranked above the constable level participated in the research, and participation in the semi-structured interviews was entirely voluntary. Because the researcher has no professional affiliation with or is employed by the police organization, the constables were recruited via an interview information sheet emailed on behalf of the researcher by the police organization. The information sheet informed the participants that their involvement in the research would be anonymous, and all responses de-identified and information such as the officer's race, ethnicity, gender, or age removed because the police organization was concerned these may identify constables working in smaller areas across the state. Whilst this information was not included in the final study, it is acknowledged that each of these variables may have shaped their interview responses. The interviews were digitally recorded and ranged between 35 and 45 min long. Each recording was transcribed into word documents for analysis, and NVivo 11 was used to facilitate the analysis of findings and the identification of key themes. The research was approved by the university's 'Human Research Ethics Committee' (approval number 1700000884).

### 2.2. Analysis of Findings, and Identification of Key Themes

A modified grounded theory analysis approach [31] was applied to analyze the findings and to determine how diversity and inclusion affect police practice. Concepts were formed from interpretations of the data using an analysis worksheet. Using grounded theory in this way enabled the researchers to apply an inductive approach, whereby theory was generated from the findings and the coding began as the interviews progressed. Key concepts were formed using an open coding method (whereby observed data and phenomena are segmented into meaningful expressions and described in a short sequence of words) and themes were categorized using a selective coding method (whereby one theme is chosen to be the core or underpinning concept which combines all other data within that group) [32]. Five core themes emerged from the interviews. This included:

(1) Broad Recognition of Diversity—Officer Characteristics, and General Identifiers
(2) Specific Recognition of Diversity—Sexuality, Gender, Nationality, and Previous Employment

(3)   Recognition and Non-Recognition of Diversity—Working Alongside Diverse Identi-fied Officers

(4)   Non-Recognition of Diversity—The Whiteness of the Organization

(5)   Recognition of Diversity—Inclusion and Exclusion in the Workplace

Analysis of the data identified clear recurring themes and that saturation of the themes was reached with 46 interviews. Once it was determined that no new information would be discovered, each of the themes was assessed in relation to diversity and inclusion and police practice. It was determined that the findings would articulate the meaningful discussion which follows the findings. In addition, the researchers were aware of their subjectivity as citizens and how this may influence their interpretation of the findings. As such, careful consideration of interpreter bias and how this may shape the meaning and interpretation of the findings was applied to the analysis, and to all the findings considered for inclusion in this research.

### 2.3. Methodological Limitations

There are several limitations to this research. First, there are limits regarding the use of semi-structured interviews because these may not be comparable to data collected in real-life situations such as during participant observation. Second, whilst semi-structured interviews were conducted with constables, interviews conducted with senior officers may offer additional insight into diversity and inclusion and how this affects police practice. We also acknowledge that the opinions of the constables in this research are not representative of all officers working in the organization, and are, therefore, not generalizable. Third, the research was conducted with only one state police organization in Australia, and as such, further research with other state police organizations across Australia could determine if these results are representative of constables (or other officers) in different police organiza-tions. Fourth, because the police organization was concerned that demographic information could identify constables working in smaller areas across the state, comparative analysis between the constable's responses and specific identifiers (such as gender) was not possible; thereby limiting further analysis of the data. Despite these limitations, this research pro-vides several important insights regarding constables and diversity and inclusion and their effect on police practice. Whilst the results are pertinent to the experience of constables employed in one Australian state police organization, the findings have relevance to other police organizations in Australia (and around the world) regarding diversity and inclusion and police practice.

### 3. Results

### 3.1. Broad Recognition of Diversity—Officer Characteristics, and General Identifiers

Across the state, there were varying responses regarding diversity and how it affects police practice. Analysis of the findings indicated that diversity represents many different things to the participants. Many of them recognized diversity as a broad quality and referred to it in terms of the characteristics they recognized, and then referred to these in identity clusters. For example, officers identified groups of people based on superordinate or face-value identity markers such as 'race', gender, and (when seen at face-value) ethnic-ity [33,34]). Subordinate identifiers (or identity markers not immediately obvious at face value) were also referred to by the participants and these included officers being identified in diverse clusters by religion, sexuality, and culture [35]. For example, Constable 25 and Constable 42 said:

*Yeah, yeah, definitely, I think we have a good diverse range of people, especially at my station, it's so diverse, like Indigenous people, different ethnicities, you know, people from different cultural backgrounds, religious backgrounds, sexualities, all that kind of stuff in such a small place.*

*(Constable 25)*

> *So, in my station that are several, like, three or four, openly like gay officers, and there's several different ethnicities, and about a quarter off officers weren't even born in Australia, a third of them are female.*

> *(Constable 42)*

Whilst it was encouraging to determine that some participants were able to define diversity broadly in relation to the existence of diverse officers working within the organization and in different police stations across the state, it was surprising that participants within this research were not able to identify different types of diverse identifiers or the nuances within the diverse categories many of the officers mentioned. For example, although some of the participants mentioned broad identifying categories such as sexuality, they referred to sexuality differences in the encompassing or umbrella term 'Gay'. They did not mention any of the different sexualities that officers could have within their workplaces such as being bisexual, asexual, pansexual, etc. Though using 'Gay' as a broad generalization to describe people with non-heterosexual sexualities is accepted by many LGBTIQ people when referring to romantic or sexual attraction between people of the same sex or gender, the term is more often used to describe men who are attracted to other men, and previous research suggests that the use of 'Gay' as an umbrella term by hyper-masculine organizations such as the police, is typically underpinned by negative bias and homophobia [34,36]. Given that in Australia, police organizations are strategically employing people who people from diverse groups such as LGBTIQ self-identified people identify as well as people from culturally diverse communities such as members of Asian, Sudanese, or Muslim communities, this finding was unexpected [34,37].

Some of the participants only recognized officer diversity in relation to observable physical differences such as differences in the age of officers working within their police station or workgroup. For example, Constable 8, and Constable 38, said:

> *We've got a lot of younger officers and we've got like, quite a few older officers and that sort of thing but it's the younger officers that are the biggest group. There seems to be like a clear difference in the groups, you know, younger officers and then older officers, so yeah diverse in that way, but I think it's normal for police to be younger.*

> *(Constable 8)*

> *In terms of age, I think that's probably where we're lacking in diversity, you know, in terms of experience, and that kind of thing, it's a very young station in terms of age, we need more older officers, but I guess policing is a young person's job.*

> *(Constable 38)*

Whilst the age of the participants was not collected (as per the ethics agreement) most of the constables participating in this research were probably aged in their late twenties with only a few constables interviewed being over the age of 35 years of age. The recognition of age difference in the workplace, therefore, particularly in relation to police work being associated with young people is interesting. The participant's acknowledgment of age in relation to the recognition of diverse groups of people within policing suggests that the constables in this research view police work in terms of an officer's capability to do the work, and age (being younger or older) is associated with an officer's capability to do the job. The responses to recognition of diversity in relation to some participants only recognizing age differences in officers suggest that police work (and capability) is underpinned by ageism and age discrimination with younger officers being seen as inexperienced but capable and older officers being seen as experienced but incapable of doing the work because as Constable 38 said *'policing is a young person's job'*.

Many of the constables in this research accepted the notion that police work is a young person's profession and yet across Australia police organizations are strategically recruiting mature age people to enter policing [38]. This is because mature age people are thought to bring 'life experience' or 'life skills' to police work which may, in turn, enhance police–citizen engagement, and many Australian police organizations have increased the

recruitment of mature citizens with a view to increasing positive community relations and decreasing citizen complaints [11]. Whilst it cannot be determined if the police organization had strategically sought to recruit younger members of the public into policing, it may be that there are higher numbers of younger aged constables in the organization.

Some of the constables only recognized the diversity in relation to different personality types working within their stations. For example, Constable 6, and Constable 40 said:

*Yes, in personality wise, and stuff like that. There's a massive diversity.*

*(Constable 6)*

*All diverse personalities, everything. The station that I'm at, I don't think there's so many officers from different cultural backgrounds, but they're all very like diverse personalities and they come from different places in Australia.*

*(Constable 40)*

Whilst a broader mix of personalities in policing may mean that better quality decision making may occur and officers are more likely to consistently perform professionally when engaging with the public [39] when some of the participants spoke about the diverse mix of personalities in the workplace, they referred to this in terms of an officer's ability to perform as a team player or in relation to their competency to complete tasks. For example, Constable 39 said:

*There's a lot of diversity. Like, you get some of the officers that are a little bit lazy, and it just kind of like, this is how I've always done it, then you get other officers that are enthusiastic, and they try and do as much as they can. And then you get other officers that seem a little bit jaded, and you have to kind of be like, why are you doing the job?*

*(Constable 39)*

Police work challenges officers to act ethically, particularly in working environments where aggressive behavior and threat are commonplace, and where police interaction occurs in hostile situations, often resulting in complex decision-making and discretional outcomes of justice [39]. Although diversity in the workplace often refers to differences in workplace approaches to problem-solving, and task completion, the connection some officers made between recognition of diversity and an officer's levels of competency and an officer's ability to work professionally is problematic, given that police officers are employed as first-responders to act in situations that require public order and safety, enforce the law, and prevent, detect, and investigate criminal activities. It is also problematic given that constables are the lowest ranked officer within police organizations in Australia and research determines that senior officers have an influence over junior officers' competency in the workplace and the way they police citizens over time [40].

*3.2. Specific Recognition of Diversity—Sexuality, Gender, Nationality, and Previous Employment*

Whilst some of the participants recognized diversity broadly in relation to general identifiers and characteristics, many of them recognized the diversity in specific ways by identifying characteristics or identities specifically associated with diverse groups. As stated, although some of the participants spoke about 'Gay' officers as a broad generalization to describe officers with non-heterosexual sexualities working in their stations, when asked about diversity in their workplace, many of the constables spoke specifically about gay (male) and lesbian (female) officers that they work alongside. Only one of the participants spoke positively about officers with non-heterosexual sexualities in policing and the impact their inclusion can have on police work. For example, Constable 18 said:

*There are gay people, and I find that to be good because these people are just normal people like others out in the world, and we're going to deal with people with similar issues, similar backgrounds, and things.*

*(Constable 18)*

The acknowledgment that gay people working in policing are reflective of the members of the community being policed reflects how well this constable is aware of their community

and the type of people within it. Research suggests, however, that officers differ in their perceptions of the communities they police and are not always aware of the minority group members within them [41]. Criticism of police from members of diverse groups stems from a lack of acknowledgment of police regarding the nuances of the communities they work with as well as differential policing of citizens due to officer bias towards diverse groups of people [34,42]. Bias towards citizens whose identity marks them as diverse or different from majority group members has been found in much of the policing literature examining police interaction with people with non-heterosexual sexualities [41]. Many of the participants in this research expressed negative recognition of diversity in relation to gay and lesbian officers and referred to homosexual officers in derogatory terms. For example, Officer 20, and Officer 46 said:

> *I'm aware of the gays but only because I've had a little bit of exposure previously with a lot of people who were sort of homosexual. But I would say no, though, as a rule, a lot of people are quite normal. and not gay at work.*
>
> *(Constable 20)*

> *I think there's more females than males in our current station. Oh, and there's a couple of homos in the station.*
>
> *(Constable 46)*

The use of derogatory language to refer to diverse people with gay or lesbian sexualities creates discrimination and disadvantage for the officers being categorized in this way. The grouping of officers as 'homos' by Constable 46 contributes to the 'othering' of these officers by heterosexual officers (typically most officers) within the organization [34,43,44]. In addition, research suggests that using a person's sexual identity as the pronoun to describe them or as a way of identifying them also contributes to 'othering', particularly when used to describe people with sexual or gender identities that do not conform to heteronormative expectations [45]. For example, Constable 20 said:

> *I think there was one "gay" in 'traffic', we all noticed him.*
>
> *(Constable 20)*

Bias towards groups of officers identified by specific diverse characteristics was also expressed by many of the officers (male and female) towards female officers within the organization or within their stations. When asked about diversity in the workplace many of the officers spoke about gender diversity and the number of females working within policing. For example, Constable 3 and Constable 9 said:

> *My first station, there wasn't that many females, I found a lot of them were on maternity leave. So, it was weird. They like joked about, like, be careful, which chairs you sit on, because half the females are pregnant, or having babies or whatever.*
>
> *(Constable 3)*

> *There are more males, but it's not too far off being 50/50, we've got quite a few females around here.*
>
> *(Constable 9)*

When asked to expand on gender diversity within the workplace, only one participant in this research mentioned transgender people and acknowledged the possibility of transgender people working in policing, but their response was not positive. For example, Constable 25 said:

> *If there was a transgender person in the team, it would be mentioned, we'd all know* [laughing], *but there's none in the police, as far as I know.*
>
> *(Constable 25)*

When the participant was asked to expand further on their comment, they explained that officers in their station talk frequently about their colleagues and that the identities of officers (particularly those that are considered diverse or known to be diverse) are often

discussed. They explained that a transgender officer would be *"noticed by other officers"* (non-transgender or cisgender officers) and that being gender-identity diverse would *"be a problem for that officer."* (Constable 25).

Research by [20,46] suggests that for many individuals, particularly police officers who work in hyper-masculine organizations, the gender binary of male and female are all that is known or understood about gender and the sex of the body, and as such, transgender people challenge the normative expectations of the outward appearance of males and females, and expectations regarding behaviors, gender performance, and competence. Police organizations, however, comprise large numbers of cisgender officers, and whilst transgender officers may be part of the workforce, transgender officers remain relatively hidden within police organizations [20] and while some transgender officers may be visible within the organization, many officers would not necessarily be aware that they were working alongside a transgender colleague. Some of the participants were also specific in their acknowledgment of diversity in relation to their awareness of the different nationalities of officers within their workforce and although many of the participants talked about this broadly, a few constables mentioned specific nationalities of officers working within their stations. For example, Constable 11, Constable 21, and Constable 27 said:

> *I've seen people who have different languages, like there's a Cantonese speaking, and there's an Indian officer but that's it, I think.*
>
> *(Constable 11)*

> *Yeah, quite a bit of variety and backgrounds. And everyone sort of brings their own flavour to the job. We do have like our Police Liaison Officers, and we've got people from different backgrounds and things. Yeah, so, we've got yeah Polish, we've had Indigenous Police Liaison Officers here as well. That sort of seems to be the main mix.*
>
> *(Constable 21)*

> *I think there's a lot of cultural diversity. With different, I guess, people with different nationalities or different backgrounds. Lots of accents. In the dayroom here, you'll hear a variety of accents.*
>
> *(Constable 27)*

To become a police officer in Australia, applicants must be Australian citizens. This means that although many police officers will have been born in Australia, a significant portion will have been born overseas and nationalized as citizens at different stages in life. Previous research suggests that police officers view themselves and other officers as a collective group (see [2,11,47]) and membership within the group (as police officers) affords them a cohesive group identity and sense of unity. Yet it was interesting that many officers recognized the diversity of origin and nationalities of officers within the organization and within their stations, and differentiated these officers by this, even though all police officers are formalized Australian citizens. In addition, many of the participants spoke negatively about the diversity of officers of international origin and nationalities and separated these officers from other Australian officers within the organization or within their stations. For example, Constable 2, and Constable 5 said:

> *Well, at my station people born outside of Australia outnumber people born in Australia. There are others, but they shouldn't be here, I mean, I know the organization wants to represent the country, and a lot of different nationalities. They want to show how "diverse" the police can be. But don't do it, just because you can, do it because they deserve to be there. Most of them don't.*
>
> *(Constable 2)*

> *There is an officer who is from India, or Sri Lanka, and yeah, some women as well, there's one or two people that aren't straight, but I don't really pay much attention to it, because, you know, it's not worth it.*
>
> *(Constable 5)*

Recognition of diversity was also discussed by some participants in relation to the diverse areas of employment some officers held before entering the police organization. Referring to diversity in this way, some participants linked diversity to an officer's ability or competency to work as a police officer. For example, Constable 10, and Constable 16 said:

*The other officers I talk to come from different walks of life, like, they've all had different jobs in the past, I think it helps with this job.*

*(Constable 10)*

*We've got people from all different backgrounds, and different previous jobs, like one was a ski instructor in Canada prior to joining. And we've got one who's a Brazilian jujitsu instructor, which is good for policing, so yeah, lots of diversity.*

*(Constable 16)*

Research suggests that police organizations across the globe are actively recruiting people who have 'life experience' and who have worked in different occupations prior to entering policing, in the view that prior experience will enable or better prepare officers to express empathy and compassion in a wide range of situations or contexts [48,49]. Life experience in relation to how this affects an individual's resilience to change as well as their ability to cope and respond in different situations is also strategically determined in recruits as they enter the police academy [50]. Prior recognition of life experience in relation to previous employment, therefore, is viewed positively in terms of it being a skill that can be used to assess the ability and capability of recruits to engage in police work [50].

*3.3. Recognition and Non-Recognition of Diversity—Working Alongside Diverse Identified Officers*

Whilst some of the officers spoke about the diversity of previous employment potentially helping an officer to engage in police work, many of the recruits spoke negatively about diversity in relation to an officer's ability and competency and working alongside diverse officers. Some participants spoke about this generally, in broad terms, but it was very clear from the analysis of these responses that officers from the majority group or identified as not being diverse clearly differentiate themselves from officers identified as belonging to a diverse group. It was also clear that many participants were unhappy with the number of diverse officers being recruited into the organization. For example, Constable 28, and Constable 40 said:

*I think they need to do less. And they need to bring in less diverse officers. You don't need to get them forced down our throat for people to accept things. And I think they just need to be careful that by accepting some people, they are pushing others away.*

*(Constable 28)*

*I feel like that they have been dropping the standards, to you know, make the "ideal" force, and for them to include, you know, "diverse" people.*

*(Constable 40)*

Some of the participants mentioned specific diverse identifiers such as gender and 'race' and spoke negatively about working alongside officers identified with these characteristics. Blaming workplace incompetence on the collective identity or attributes of individuals such as gender or 'race' impacts the behavioral reactions of those whose identities are blamed as well as the behaviors of those who identify the cause [19]. Many studies have examined the effects of the blame on majority and minority group members from each perspective (see [51,52]). Whilst outcomes of blame affect each group differently, blaming diverse groups of people within a collective group comprising majority and minority identified people, strengthens the identities of majority group identified people (thereby facilitating group-based control), and enhances the individual and group perceptions of dominance and superior status of majority group identified people within the group [19].

Measures of prejudice or bias towards groups of people are frequently based on the collective characteristics of the group being blamed. This is usually underpinned by a

loss of control over a situation; the less control a person has in a situation the more likely they are to apportion the blame to a superordinate identifier (such as gender or 'race') or subordinate identifier (such as sexuality or religion) as the cause of the problem, particularly if there is no other factor or reason for loss of control [46]. This is interesting, given that police officers are all trained to respond and act the same way, they must follow the same guidelines, and have the same tools and powers, and responsibilities. Yet some constables in this study blamed the diversity of officers as being the cause of negative workplace experiences. For example, Constable 22, and Constable 42 said:

> *They were taking as many females as they could but whether they were good enough for the job or qualified enough, who knows. And some 'shit' happened, and I heard the mistakes that some of those officers did, and now we're stuck with them.*
>
> *(Constable 22)*

> *We turned up to a job, people were drunk, and agitated, and there's him, the Indian, trying to tell them what to do. They just wouldn't accept it, you know, because he's Indian. It caused me a lot of problems.*
>
> *(Constable 42)*

Some participants spoke negatively about diversity in the workplace in relation to working alongside gay-identified officers. For example, Constable 2 said:

> *They are many officers who are not accepting of gays, and I'm just like, I understand, you know, I'm Catholic. And Catholics are all, you know, abortions and gays are bad.*
>
> *(Constable 2)*

Many of the participants also spoke about the banter or mockery that is expressed in the workplace towards gay male officers, and that the banter is a normative part of policing practice and police workplace culture. For example, Constable 3, Constable 10, and Constable 46 said:

> *There was one gay officer, a gay guy, and it was talked about, in a bad way and not in a light-hearted way, it was talked about until he transferred, it is still talked about, joked about, but just banter you know.*
>
> *(Constable 3)*

> *It sort of depends, I know that within our team we do banter about the gay people but it's not in a homophobic way, it's like, "did you see how good looking that guy was" to the gay guy in our team, who is gay, and we say, "Oh, bet you'd like to go there?".*
>
> *(Constable 10)*

> *We have a few jokes with them, but it's all in good fun you know, about them being poofs and that.*
>
> *(Constable 46)*

According to [29], this type of talk or "canteen banter" is common within police organizations, and whilst research suggests that police officers engage in banter to make light of highly stressful situations when banter is frequently used by officers to refer to groups of people, or features people's identities, it is underpinned by bias and prejudice [4]. Whilst the responses from the participants were predominantly negative in terms of working alongside diverse officers (particularly gay officers), some of the participants expressed positive opinions about working alongside diverse officers and acknowledged the benefits of having diverse colleagues in relation to conducting police work. For example, Constable 9, and Constable 24 said:

> *Females are better communicators, I think, which is good, good to have someone that's a good communicator, working with you. makes the job a lot easier, I get better results, we can get things done quicker, to a higher standard.*
>
> *(Constable 9)*

*Like in our station, the most loved person is our Indigenous liaison officer, he can't go anywhere without a thousand people going, "Hey", even though we're not disliked in our community, they will tell him more than they'll tell us, because he's one of 'them'.*

*(Constable 24)*

Many participants expressed that they do not identify with the diverse officers that they work alongside and felt that they were being pressured by the organization to identify with diverse officers. Some participants also said that had problems associating diverse officers as fellow police officers, in the same way, they would non-diverse officers. For example, Constable 5, and Constable 21 said:

*Because you're white, and middle aged, now we're this group that must sit down and shut up. Now it's because anything we say, is now seen to be like, racist or ignorant, or this and that, so now we must "identify" with others, even though we don't, and don't want to, not really.*

*(Constable 5)*

*It's still very new to me, the diversity, like even when little kids come up and are excited and the mum says, "Hey it's a police officer", and points to one of them who's not Australian, who's not White, it's still kind of like, 'oh, yeah, that's right'.*

*(Constable 21)*

This finding is interesting given that officers usually identify as a collective group (regardless of specific identifiers) when referring to themselves and present a unified front to citizens in terms of being 'police' [42]. It was also interesting that some officers specifically mentioned identifying with characteristics present in other officers within their stations. For example, Constable 16 and Constable 33 said:

*I identify with the male officers. All of them are heterosexual, so yeah, I do identify with them, but I wouldn't identify with the diverse cops, no.*

*(Constable 16)*

*I'm normal, heterosexual, so I identify with people like me, not the others. It's like I am part of a little club, while you're an officer, you're part of a little club, or maybe a massive club.*

*(Constable 33)*

Not identifying with diverse officers, distinguishes and strengthens the identities of non-diverse officers to diverse and non-diverse officers, particularly if non-diverse officers comprise the majority group within a station or across the organization. Like the association the participants made between a diverse officer's identity and notions of competence, purposefully not identifying with diverse officers facilitates group-based control, particularly for non-diverse officers within an organization like policing, that is based on heteronormative, hegemonic masculinity and subsequent notions of behavior [3]. The cognizant non-identification with diversity, and non-identification with other diverse officers, enhance notions of dominance and perceptions of the superiority of non-diverse officers towards diverse officers, which creates additional barriers within the workplace and between groups of people [19].

### 3.4. Non-Recognition of Diversity—The Whiteness of the Organization

The perceived barriers between diverse and non-diverse officers were also upheld by many of the participants who did not recognize diversity and only recognized the 'Whiteness' of the organization. Research determines that Whiteness is a status that confers unequal social, political, and economic freedoms across multicultural and multiracial societies [53], and whilst some argue that Whiteness cannot be reduced to specifics of race, it is also used as a collective grouping to refer to organizations or institutions that are dominated by White people [54]. The reality of policing is that most Western police organizations are dominated by numbers of officers who identify as White, as well as being

dominated by officers who are White, male, and Catholic [34,55] Most police organizations (particularly in Australia) comprise smaller numbers of officers from diverse groups and many of the participants in this research recognized this in relation to the stations where they work as well as across the police organization. For example, Constable 22, and Constable 45 said:

> *I guess the police force is a brotherhood. White guys, early 40s, tall. That's what most Cops are.*

> *(Constable 22)*

> *We're all white in my station, all male, one female, all normal, you know.*

> *(Constable 45)*

Research suggests that most White people generally consider their race to be irrelevant to their actions and perspectives on the world, whilst at the same time using 'Whiteness' as an implicit comparison regarding acceptable notions (and assumptions) of behavior (see [53,54,56,57]). Yet some of the officers in this research were highly cognizant of the Whiteness of the organization and of the Whiteness of other officers and expressed this in relation to heteronormativity. For example, Constable 24, and Constable 29 said:

> *Predominantly white, and I still think, predominantly heterosexual. Yeah, I would say that I could only name on one hand, the number of homos that I've come across.*

> *(Constable 24)*

> *The police? it's a boy's club. It is very white and straight. Most officers are Australian and White, that's it.*

> *(Constable 29)*

Whilst this paper does not have the scope to go into the complexities of Whiteness and racial awareness amongst police officers, it is important to note the issue of Whiteness in relation to police engagement because the challenge of interaction with diverse members of society may heighten the awareness of 'Whiteness' of some officers; particularly for those officers that do not recognize diversity, or for those officers who are challenged by engaging with or working alongside diverse people and officers. This idea is supported by [58] who argue that when a work environment fails to support a diverse workforce, and one identified group dominates the organization, negative outcomes such as an increase in harassment, discrimination, and intergroup conflicts will occur; particularly if members of an organization do not identify with, or understand, the needs of another group they are charged with helping. It is also likely to increase barriers within the workplace that will exclude diverse officers from belonging to the group and feeling included [11].

*3.5. Recognition of Diversity—Inclusion and Exclusion in the Workplace*

Critics argue that workplace diversity has the potential for positive and negative outcomes regarding inclusion in the workplace [58]. When diversity is well managed, and the workplace supports and values its diverse personnel, employees from all backgrounds will feel included within the organization [58]. Policing research, however, suggests that inclusion in the workplace for diverse officers is complex and subject to the policies and workplace practices espoused by the organization [59]. Some of the participants in this research spoke about the organizational exclusion of diverse officers within the workplace. For example, Constable 25 said:

> *You hear stories about people who are diverse putting in grievances who have pretty good reasons, and they're not getting accepted, and it does make you wonder why, what is the real reason they are not being treated the same as other officers.*

> *(Constable 35)*

Other participants spoke about recognition of diversity in the workplace giving advantage to diverse officers in terms of increased inclusion within the organization as well as additional support from management and senior officers regarding leniency in terms of

police work and leniency when diverse officers make mistakes. For example, Constable 22 said:

*I don't know, I'm kind of neutral on that one, I think diverse people get more inclusion or get more stuff than other officers, it's like they are treated better I think, they get more leniency.*

(Constable 22)

Research suggests that diversity in the workplace is often perceived as a threat to equal employment opportunity, workplace fairness, and equal treatment of all employees by non-diverse people whose misperceived notions of diversity in relation to workplace advantage heightens their overall perception of unfair treatment; particularly when comparing themselves to diverse employees [60]. The negative effect of this misperception results in unbalanced status and power relationships between employees and within workgroups [61]. Diversity as a factor giving diverse police officers an unfair advantage in the workplace is an interesting notion and one that is still in need of further inquiry in policing research. One Australian study by [7] argues that senior police officers (such as police detectives) have negative perceptions of diversity, and associate diversity with unfair advantages in the workplace, which creates barriers to inclusion for diverse officers.

Much of the international body of research examining diversity in policing posits that being identified as a diverse officer increases the likelihood of exclusion from the collective group by other officers [47,59,62–64]. Critics of police organizations argue that this is a long-standing problem in policing and the lack of inclusion of diverse officers needs to be addressed within the culture of policing [63,64]. Many of the officers in this research also spoke about the exclusion of diverse officers being affected by negative associations that non-diverse officers make between police work and engagement with diverse groups. Some of the officers specifically mentioned the link non-diverse officers make between frequent police engagement with citizens from diverse groups as well as negative aspects of police work and policing of diverse people. For example, Constable 19 said:

*Can I be honest? Then no, diverse officers are probably not included, not 100 percent. Being diverse and being included, don't go together in policing, maybe it's because cops feel the need to distance themselves from the people they're dealing with, and maybe they see the diverse officers as being part of those groups.*

(Constable 19)

Although police engage with all members of society, the biased association officers make between negative aspects of police work and diverse citizens is problematic given that they are meant to police all citizens fairly and is also problematic given the connection officers then make with this biased association and diverse officers within their team or organization. Police work requires officers to work as a team in situations of heightened stress, aggression, and disorder, as well as being unified in their response to highly complex criminal situations, which require officers to work collegiately and professionally with one another. If officers are excluding colleagues based on diversity, then it stands to reason that these officers are less likely to police citizens with high ethical standards and will be less aware of the accountability of their actions, and will be inconsistent, unfair, and dishonest in the performance of their duties [2]. Exclusion in the workplace of diverse police officers, therefore, is not only going to cause problems for the officers who are unable or unwilling to work with diverse officers as a cohesive team, but it is also going to cause problems for the competency of police organizations to resolve problematic situations between different groups of people.

## 4. Discussion and Conclusions

The othering of diversity and diverse officers by constables within the police organization in this research is problematic given that constables are in the early stages of their policing career and are taught at the academy that unification, teamwork, and solidarity underpin policing. The lack of unification constables in this research have with diverse

colleagues is also concerning given that a cohesive police team increases the safety of all officers, improves the effectiveness of police response, strengthens the communication between police and citizens (as well as communication within the organization), increases the morale of officers, and will support the legitimacy of the organization [3,5,6]. The bias constables expressed towards recognition of diversity was overwhelmingly negative and constables in this research are recalcitrant to work with diverse officers who they believe are not competent and who are perceived to be unfairly supported within the organization. The diminished positivity constables in this research have towards working alongside diverse officers is surprising because research suggests that lack of enthusiasm for policing and police work generally diminishes over time as policing careers unfold, not in the early stages of a police career [14,27–30].

Most constables enter policing with high-minded notions of integrity regarding police work and being part of the team, yet the level of cynicism and negativity constables expressed towards diversity and diverse officers will certainly affect the collective sense of unity expected within a policing team. Whilst constables in this research were not asked questions about their own implicit or explicit levels of bias towards members of diverse groups, the unsolicited responses from many of the constables, as well as the recognition of Whiteness in terms of the racial identity of many officers within the organization suggests that constables in this research are biased towards officers that are not part of the majority group. The cognizant awareness and recognition of the dominance of police officers who identify as White by constables in this study are interesting given that research posits that Whiteness is normally overlooked by White people in terms of the conscious awareness this has on a person's worldview [57] and that it is often difficult for members of a majority group to see beyond their own perspective. In this research, however, the constables' cognizant recognition of Whiteness is not just about recognition of the color of the officers' skin, it is also about the ideologies and attitudes towards diversity and the way it causes perceived or actual inequitable distributions of power [57]. Recognition of the presence of a dominant race (in this instance, the large number of white identified officers within the organization) not only represents the dominant group but is also a conscious (and unconscious) recognition and acknowledgment that police organizations were historically, and are currently, structured around the dominant group comprising majority group members [57]. It is also recognized that there is an opposing group to the dominant group and this group is categorized under concepts and characteristics of diversity and labeled 'diverse'.

Numerous pieces of research suggest that whilst police officers share a collective identity under the label 'police', this group identity is only relevant when being projected outward toward citizens [3,11,41,46,65–67]. The outward projection of a collective group identity enables police to differentiate themselves from citizens, thereby enabling police to categorize non-group members either by their non-employment as police officers or (in the case of many diverse groups of people) classify civilians as outgroup members due to differences in identity, which mark them as being different to majority group members [3,11,46,66]. The collective police identity, therefore, significantly shapes officers' policing of groups of people with whom they do not identify [3,46,68–70]. Within the police organization, the collective group identity also enables the organization to maintain rules, procedures, and the hierarchical management of the organization. Yet research suggests that the collective group identity shared by police is not completely cohesive [3,46,71]. It is argued that the group identity is segmented, with the larger group identity comprising smaller groups with officers collectively grouped based on differences in rank, gender, race, and other characteristics [71]. Just as officers differentiate citizens by identity, the constables in this study also differentiate colleagues by identity and reject differences considered diverse.

Although national diversity and inclusion strategies offer Australian police organizations the opportunity (and resources) to support diverse employees, the reality of policing in the twenty-first century is that most Australian officers identify as White, heterosexual,

and male [3]. In policing, organizational policies are meant to increase the sense of inclusion diverse officers feel within the workplace, as well as set standards regarding workplace performance. Diversity and inclusion strategies are also meant to help implement or support a workplace culture in policing that minimizes discrimination, harassment, victimization, and bullying [72]. Research suggests that police organizations are beset with discriminatory workplace problems and practices, and that policing as a profession is dominated by outdated notions of police work and the attributes or characteristics individuals need to be competent and capable officers [72]. Whilst many police organizations have reevaluated traditional notions of recruitment regarding officers, police work is still dominated by notions of machismo, underpinned by hegemonic masculinity and exaggerated forms of masculinity.

Traditional recruitment practices in Australia operationalized the ideal officer, and whilst modern Australian police organizations have focused on overcoming these outdated ideas of the idealized officer, many of them still base their guidelines on conventional ideas regarding what police work entails and how it should be conducted [3]. However, much of the body of the literature examining police work determines that positive policing and outcomes of justice, as well as positive police–citizen engagement, relies more on the inclusion of diverse people within policing and that this inclusion will diminish outdated traditional modes regarding how police conduct work [3,46,68,70,71].

Yet, the constables in this study uphold barriers to inclusion because of their negative recognition of diversity and subsequent rejection of it. By labeling officers as diverse, constables in this research reject officers identified in this way, which may have a detrimental effect on inclusion and serves to increase isolation between the diverse officer and colleagues [3,9]. The rejection of diverse officers may also be supported by the public's perception that the police are a powerful (and masculine) organization [3]. This idea begins at the academy and is instilled in recruits at the start of their police career and excludes any behavior or person not considered appropriate or the ideal officer. The conscious and unconscious bias and stereotypes constables in this research have towards diversity as well as the stereotypes linked to diverse officers and their competency to conduct police work have a detrimental effect on the capacity of diverse officers to do their job and be included within the police organization. Underpinning much of the barriers to inclusion of diverse people in policing is the association constables make between the characters of diverse citizens being policed and the characters of diverse officers they work alongside. This is highly problematic given that the constables are associating criminal behavior with specific superordinate or subordinate attributes and identifiers, and police are already subject to extensive criticism regarding their tendency to link criminal behavior with certain groups [69].

Research indicates that there are significant differences in awareness levels of officers regarding the rank of officer and working with or engaging professionally with members of diverse groups [11,46,71]. Whilst many police organizations will most likely screen out applicants to policing who display evident prejudice towards diverse communities, subtle and socially driven stereotypes regarding working alongside or policing diverse people are more difficult to detect [3,11,34]. Police organizations are therefore conscientiously recruiting officers who display an ability to interact and engage with diverse groups of people [3,11,34]. Recruitment of the 'right sort of person' however, does not guarantee the ability of an officer to work alongside members of diverse groups and does not diminish the importance of the organization in its role to reduce workplace bias toward diverse people or to encourage the inclusion of all employees [71].

Although it is understood that the inability of police organizations to create inclusive workplaces will have an impact on workplace performance [59], it is not understood how the lack of inclusion in the workplace will affect the work-related behavior of officers in terms of how they perform their role whilst working alongside colleagues who are not included. In Australia, most police officers usually work ten-hour shifts (apart from New South Wales police officers who generally work twelve-hour shifts) and respond to calls for

assistance and incidents of crime with one other officer or shift partner. As such, they must rely on their partner to support and protect each other during threatening or dangerous situations. The sense of comradery and partnership that inclusion supports, is, therefore, an important component underpinning an officer's sense of security when working alongside their shift partner.

The insecurity of not being included by colleagues and the ostracism this creates is detrimental to the ethos of the policing response, which requires officers to co-respond and to work in partnership. Facilitating an inclusive culture within police organizations is not impossible, but it requires careful planning and input from the organization as well as employees. Inclusion within a workplace is based on a cognizant recognition by all employees, that all colleagues are accepted and included in the workplace and have access to the same information, resources, and support available for all employees, are involved in work groups and are able to participate in decision-making processes, and who are accepted and receive fair treatment which positions them as an insider within the organization [73]. Inclusion of diverse officers within a police organization will certainly increase the level of cohesion experienced by all officers but first senior officers and police management must take initiatives that will reduce officer bias, discrimination, and prejudice towards diverse officers and lessen negative perceptions regarding diversity and workplace competency. Developing organizational practices to reduce diversity bias and increase inclusion will enable senior officers and police management to develop inter-reliant teams that are confident in the team's response to all policing situations.

Workplace practices and policies in Australian police organizations are based on fairness and equal opportunity, yet many organizations are criticized for not effectively responding to discriminatory policing practices, unfair and biased workplace practices, and poor interpersonal treatment of officers [7]. This research determines that lack of acceptance of diversity broadly, and bias towards diverse identified officers, results in the exclusion of officers and a workforce that is fragmented because officers are unlikely to work collegially. The lack of inclusion of diverse officers has implications for workplace competency and the inability of officers as a collective group to police equitably. Inclusion of all officers within police organizations is likely to increase the mental capacity of officers and reduce workplace stressors [74]. Lack of inclusion, therefore, may increase the likelihood that officers rejected by diverse identification may result in poor workplace performance, differential policing of citizens, and attrition from policing [7,75]. If the inclusion of all officers is to be effective, police organizations must target diversity bias and address officer wellbeing, thereby creating (and supporting) an inclusive work environment.

**Author Contributions:** Conceptualization, T.M.-J. and S.F.; methodology, T.M.-J. and S.F.; formal analysis, T.M.-J. and S.F.; writing—original draft preparation, T.M.-J., writing—review and editing, S.F. All authors have read and agreed to the published version of the manuscript.

**Funding:** This research received no external funding.

**Institutional Review Board Statement:** This study was conducted in accordance with the guidelines of the National Statement on Ethical Conduct in Human Research (2007—Updated 2018), and Queensland University of Technology Human Research Ethics Committee Approval: 1700000884. Approval date 25/10/2017-25/10/2023.

**Informed Consent Statement:** Informed consent was obtained from all subjects involved in the study.

**Data Availability Statement:** The data presented in this study are available on request from the corresponding author. The data are not publicly available due to ethics protocols.

**Conflicts of Interest:** The authors declare no conflict of interest.

## Appendix A. Interview Prompts—Points for Discussion

1.  How would you describe your role as a police officer?
2.  How would you describe the police?
3.  Can you tell me how you see the police as an organization?

4. Do you identify strongly with other police officers?
5. Are there any officers that you don't identify with?
6. Can you give me an example of how you think you identify with other police officers?
7. Can you give me an example of how you think you don't identify with other police officers?
8. How important is it to you—to belong to the police?
9. Is being a police officer important to your identity?
10. What does diversity mean to you?
11. What do you think diversity means to the organization?
12. Can you describe the diversity you see across the organization?
13. What do you think the organization could do to increase the diversity of its workforce?
14. Can you describe the diversity you see within your police station?
15. Does the diversity of an officer affect their workplace performance?
16. Can you describe the effect this has on their workplace performance?
17. Can you describe the effect diversity has on overall workplace performance?
18. What does inclusion mean to you?
19. What does inclusion within the organization mean to you?
20. Does the organization include all its officers?
21. What do you think are the factors that could exclude an officer?
22. What could the organization do to include all its officers?

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
