# Peer review of "“Being Diverse and Being Included, Don’t Go Together in Policing”—Diversity, Inclusion, and Australian Constables"

_societies, doi:10.3390/soc12040100_

Round 1

Reviewer 1 Report

This is an important paper, which sets on to understand the role of diversity in police organizations in Australia. The paper is very well written and organized and easy to follow. The research is based on original research of conducting semi-structured interviews with 46 respondents. The topic, as well as the conclusions are very important, and could lead to a productive discussion. 

I have three recommendations for the author. The biggest issue in my mind is that the author had to forgo all markers of diversity for the respondents. This has to be addressed in the paper – how does the lack of this knowledge about the respondents affect the results? Much more would have been learned if we knew the gender, race, ethnicity, etc. of the respondents. 

In addition to addressing explicitly the issue and think about the consequences for the results, one other remedy is to find objective statistical information about the composition of the police organizations that the author studies. Including a table with the police force characteristics, studied in the article, will be important (such as gender, sexual orientation, race/ethnicity, age, etc.). 

The second issue is that sometimes the author makes quite pronounced generalizations based on a few dozen interviews. For example, in order to accuse an organization of ageism or age discrimination a lot more evidence is needed than a few remarks by interviewees. I would suggest that the author considers that results from the interviews a bit more carefully, particularly since this is not a representative sample of the police force. 

Author Response

Reviewer 1 suggestion 1 - thank you for this suggestion - whilst page 3 lines 141-143 states 'the officer’s race, ethnicity, gender, or age removed because the police organization was concerned these may identify constables working in smaller areas across the state' - we have also added information to the limitations of the research (page 4 lines 187-190) acknowledging the limitations of the lack of this information in the research - 'Fourth, because the police organisation was concerned that demographic information could identify constables working in smaller areas across the state, comparative analysis between the constable’s responses and specific identifiers (such as gender) was not possible; thereby limiting further analysis of the data'

Reviewer 1 suggestion 2 - thank you for this suggestion - But, if we add 'objective statistical information about the composition of the police organization' - it will more than likely identify it - there are only 8 state organisations - as such this is not possible but we have addressed the issue - see suggestion 1.

Reviewer 1 suggestion 3 - thank you for this suggestion - however, we are not accusing the organisation of ageism - we have checked the article and it is not doing this - we do acknowledge that the views of the constables are not generalisable or necessarily representative of the organisation and we do address this in the limitations - for example, it says 'Second, whilst semi-structured interviews were conducted with constables, interviews conducted with senior officers may offer additional insight into diversity and inclusion and how this affects police practice. We also acknowledge that the opinions of the constables in this research are not representative of all officers working in the organization, and are therefore, not generalizable.'

We thank the reviewer for their suggestions and for taking the time to respond to us. Thank you. 

Reviewer 2 Report

I think that the article is very interesting. However, before the final publication, I would suggest making a couple of revisions:

1 - I wonder if the empirical results of the research based on the responses of 46 constables are representative of the whole community of Australian policemen. Moreover, these interviews were conducted only in one state and were semi-structured. I would recommend the author to narrow down the scope of the research project and be a little bit more specific with the geographical and socio-cultural indicators. I wonder if the results presented in the paper are relevant to some local community, or if they may also be representative of the whole country. I would be curious to know what the international community of policemen could learn from your research project as well.

2 - I think that the qualitative analysis of the research results is very comprehensive. At the same time, I wonder if it would be possible to include some statistics. For people who are not experts in the topic it might be interesting to see some figures and numerical considerations. For example, how would you measure bias, recognition, or diversity? 

3 - In my opinion, the 'discussion and conclusion' section is too long. I would suggest separating it into two independent sections, 'discussion' and 'conclusion'. In Discussion, I would summarize the findings of the research project. In Conclusion, I would discuss the significance of your research findings for the current academic debate on the issue and consider the practical applications of your research project (I would try to give policy recommendations) in Australia and abroad.

Author Response

Reviewer 2 - Suggestion 1 - we thank the reviewer for this suggestion - we have expanded the limitations section to address the issue of representative data - the limitations now states from point 2 raised -  'Second, whilst semi-structured interviews were conducted with constables, interviews conducted with senior officers may offer additional insight into diversity and inclusion and how this affects police practice. We also acknowledge that the opinions of the constables in this research are not representative of all officers working in the organization, and are therefore, not generalizable. Third, the research was conducted with only one state police organization in Australia, and as such, further research with other state police organizations across Australia could determine if these results are representative of constables (or other officers) in different police organizations.'

Given the nature of the Australian state police organization - we cannot specify or narrow down geographical indicators because this would likely identify it - which goes against the ethics agreement. 

We also acknowledge in the methodological limitations that - 'Whilst the results are pertinent to the experience of constables employed in one Australian state police organization, the findings have relevance to other police organizations in Australia (and around the world) regarding diversity and inclusion and police practice.' In addition, the findings are situated within national and international discussion and are therefore, highly relevant to national and international police agencies.

Reviewer 2 - Suggestion 2 - we thank the reviewer for this suggestion - but this is not a statistical piece and there are no statistics that we can offer with this research.

The issue of the constables bias is addressed on page 8 line 342 and in the discussion lines 460 and line 497 and line 637 and throughout the discussion/conclusion.

Reviewer 2 - Suggestion 3 - we thank the reviewer for this suggestion - but after careful consideration of their idea we have decided to leave it as it is because we feel that the discussion of the findings directly after each theme gives the findings more depth and contextual understanding. Whilst we agree with the suggestion that policy recommendations could be made, the emphasis in this paper is on police practice, not policy change, and as such the article addresses the problematic nature that diversity and inclusion has on policing practice.